# Sequential Immune Acquisition of Monoclonal Antibodies Enhances Phagocytosis of *Acinetobacter baumannii* by Recognizing ATP Synthase

**DOI:** 10.3390/vaccines12101120

**Published:** 2024-09-29

**Authors:** Dong Huang, Zhujun Zeng, Zhuolin Li, Mengjun Li, Linlin Zhai, Yuhao Lin, Rui Xu, Jiuxin Qu, Bao Zhang, Wei Zhao, Chenguang Shen

**Affiliations:** 1BSL-3 Laboratory (Guangdong), Guangdong Provincial Key Laboratory of Tropical Disease Research, School of Public Health, Department of Laboratory Medicine, Zhujiang Hospital, Southern Medical University, Guangzhou 510515, China; hd645289511@163.com (D.H.); 13246386949@163.com (Z.Z.); lizhl1013@163.com (Z.L.); limengjun1625@163.com (M.L.); zll15649856209@163.com (L.Z.); lyh2001213@163.com (Y.L.); zhangb@smu.edu.cn (B.Z.); zhaowei@smu.edu.cn (W.Z.); 2Medical Laboratory Department, Guangdong Provincial Hospital of Chinese Medicine, Zhuhai 519015, China; xuruiyy2008@126.com; 3Department of Clinical Laboratory, Shenzhen Third People’s Hospital, National Clinical Research Center for Infectious Diseases, The Second Affiliated Hospital of Southern University of Science and Technology, Shenzhen 518112, China; qujiuxin@163.com; 4Key Laboratory of Infectious Diseases Research in South China, Ministry of Education, Southern Medical University, Guangzhou 510515, China

**Keywords:** *Acinetobacter baumannii*, multi-drug resistant, pan-drug resistant, carbapenem resistant, monoclonal antibody, phagocytosis

## Abstract

**Objectives**: The aim of this study was to prepare monoclonal antibodies (mAbs) that broadly target *Acinetobacter baumannii* and protect against infection by multi-drug-resistant (MDR) *A. baumannii* from different sources. **Methods**: mAb 8E6 and mAb 1B5 were prepared by sequentially immunizing mice with a sublethal inoculation of three heterogeneous serotypes of pan-drug-resistant (PDR) *A. baumannii*, ST-208, ST-195, and ST-229. **Results**: The cross-recognition of heterogeneous bacteria (n = 13) by two mAbs and potential targets was verified, and the in vitro antibacterial efficacy of mAbs was assessed. The median killing rate of mAb 8E6 against *A. baumannii* in the presence of complement and dHL-60 cells was found to be 61.51%, while that of mAb 1B5 was 41.96%. When only dHL-60 cells were present, the killing rate of mAb 8E6 was 65.73%, while that of mAb 1B5 was 69.93%. We found that mAb 8E6 and mAb 1B5 broadly targeted MDR *A. baumannii* on the ATP synthase complex and were equipped with an antibacterial killing ability by enhancing the innate immune bacteriolytic effect of ST-208 and ST-195 strains. Both monoclonal antibodies were validated to protect against respiratory infection at 4 and 24 h via enhancing the release of innate immune substances and inflammatory cytokines, effectively shortening the disease period in mice. **Conclusions**: mAb 8E6 and mAb 1B5 significantly enhanced the opsonization process of phagocytosis against *A. baumannii* strains prevalent in southern China by targeting ATP synthase antigens thereof, resulting in protective effects in mice.

## 1. Introduction

The resistance of bacteria to antimicrobial drugs continues to pose a significant threat to public health worldwide. It is estimated that 4.3 million antimicrobial-resistant (AMR) infections occur annually in the 34 OECD countries and EU/EEA countries included in the analysis. While approximately 35% of resistant infections are acquired in healthcare settings each year across all countries included in the OECD analysis, these infections account for approximately 62% to 73% of AMR-related deaths [1].

*Acinetobacter baumannii* (*A. baumannii*), a priority critical pathogen according to the World Health Organization, has developed increased antibiotic resistance through intrinsic and acquired mechanisms, and effective treatments against *A. baumannii* infections are becoming limited [2]. Antibiotic-drug-resistant bacterial infections increased during the COVID-19 pandemic in the United States, with an overall infection rate of 35% in 2020 compared with the previous year, and more than 10,000 deaths per year were due to *A. baumannii* infection [3]. Therefore, there is an urgent need to explore effective alternatives for complex antibiotic combination therapies.

Monoclonal antibody therapy has excellent protective and preventive efficacy against severe *A. baumannii* infections, improving the immune recognition and clearance of bacteria by targeting surface antigens. However, capsular polysaccharides (CPSs) on the bacterial surface may shield cross-immunity through the enfold-conserved epitope. For example, the anti-outer-membrane protein A mAb may block adherence to the bacterial surface [4]. In addition, CPSs protect *A. baumannii* from complement- and neutrophil-phagocyte-mediated antibacterial responses, inhibit opsonization with C3b/iC3b in the complement activation pathway, and prevent membrane attack complex (MAC)-mediated bacterial killing [5,6]. The formation of MAC on the bacterial surface may be not directly related to the serum resistance of variations between *A. baumannii* strains, resulting in different susceptibilities of *A. baumannii* strains to the complement system.

A multitude of vaccine strategies have been employed to investigate mAbs against *A. baumannii* infection. These encompass the design of subunit vaccines that target specific antigenic components of the bacterium, the formulation of whole-cell vaccines that utilize inactivated or attenuated whole bacteria, and the cutting-edge application of bioinformatics to explore antigenic potential through advanced computational modeling [7]. mAb 13D6 targets K1 CPSs and binds 13% of the clinical isolates of *A. baumannii* obtained from diverse institutions in the US [8]. mAb C8 [9] and mAb 65 [10] act against *A. baumannii* by targeting single CPSs, binding 27% and 23% of >300 diverse clinical *A. baumannii* strains, respectively, and exhibiting more than half of the different strain binding ranges. The bispecific mAb C73 [11], constructed using mAb C8 and mAb 65, exhibits a better binding range of 36% compared to the original. The anti-SK44 CPS mAb binds 62%. Clinical isolate *A. baumannii* isolates (342/554) [12]. Pse-MAB1 is a mAb that specifically targets pseudoamino acids, which are found on the cell surface of *A. baumannii*, eliminating the pathogen independently of any components of the immune system [13]. The provision of the adequate range protection of cross-diversity *A. baumannii* infection through immunization with individual antigens requires further investigation. Sequential immunological strategy, based on whole-cell vaccines by inoculating pathogens that can stimulate responses to multiple antigens across subtypes, is essential to provide protection against a wide range of strains within a bacterial species. This strategy has been verified in vaccine preparation for HIV [14], influenza virus [15], and Neisseria meningitidis [16] to specifically induce a broadly cross-recognition response or neutralizing activity, and it is one of the promising potential strategies to address diverse *A. baumannii* infections.

Sequential inoculation with heterologous live *A. baumannii* produces an antiserum that cross-protects against *A. baumannii* [17]. Antiserum obtained from the inoculation of different isolates or multi-CPS genotypes of live *A. baumannii* can broadly recognize *A. baumannii*, producing antibodies that target multiple conserved protein antigens between diverse strains. This cross-immunization response does not occur when inoculating a single CPS genotype or serotype of *A. baumannii*. Therefore, choosing the appropriate target serotype or CPS genotype of *A. baumannii* is crucial to achieve broad cross-immunization. The exploration of the mechanisms of cross-protective potency mechanisms against *A. baumannii* is still ongoing. These findings lead to the unique advantage of a sequential immunization strategy to outcome high specificity of CPS in *A. baumannii*, which may be crucial to activate the classical pathway and promote the recognition of MDR *A. baumannii* by the immune system.

We, therefore, aimed to produce a batch of mAbs that broadly against the most prevalent multidrug-resistant strains in China. Based on molecular epidemiological studies, the difference of conserved housekeeping genes in ST-208 and ST-195 (Oxford scheme) mainly concentrate on a single GPI locus and classify both STs into clonal complexes 92 (CC92) or global clonal 2 (GC2), which are predominant lineages disseminated worldwide. At the same time, ST-208 and ST-195 are leading STs in most of China [18,19,20,21]. According to cgMLST blast-related alleles of evolutionary relatedness between different laboratories, non-cc92 clonal complex isolates have more allelic differences, which include ST-229 with 2259 allelic differences. This is the same as the comparison of multi-locus sequence typing (MLST), indicating low genetic similarity to related CC92 isolates [22]. ST-195’s evolutionary relatedness has been analyzed by cgMLST, revealing a high degree of sequence similarity between the genomes of intercontinental isolates and those of China [23]. The ST-208 strains isolated from China overlap with the most prevalent ST-208 strains worldwide in a sizable proportion of diversity [24]. The effectiveness of the vaccine was notably reduced when individuals vaccinated with a single antigen were exposed to a homology variant pathogen, indicating that the unique characteristics of a single immunogen present obstacles to attaining comprehensive vaccine coverage and maintaining vaccine effectiveness over time [25]. To prevent the potential evasion of antibody-mediated antibacterial activity due to variations, the low-homology ST-229 *A. baumannii* is included in immunizations with ST-208 and ST-195 strains.

## 2. Materials and Methods

### 2.1. MLST Serotype and Virulence Gene Assignment of A. baumannii

*A. baumannii* strains were obtained from two hospitals in southern China. SZ-AB57, SZ-AB61, and SZ-AB22 were isolated from patient samples at the Third People’s Hospital of Shenzhen, and ZH-AB74, ZH-AB64, ZH-AB59, ZH-AB91, ZH-AB23, ZH-AB323, ZH-AB01, ZH-AB10, ZH-AB36, and ZH-AB66 were obtained from the Guangdong Provincial Hospital of Chinese Medicine, Zhuhai. All strains were identified utilizing an automated microbial identification instrument, followed by a drug susceptibility test conducted in accordance with the guidelines outlined in the Clinical and Laboratory Standards Institute (CLSI) M100.

Total DNA was extracted from the *A. baumannii* isolates by boiling for 10 min or using a TIANprep Mini Plasmid Kit (TIANGEN, Beijing, China, cat: DP103). Specific primers were used according to a previous report [26]. Green Taq Mix (Vazyme, Nanjing, China, cat: P131-01) was added to increase the sample volume to 50 µL, and the samples were amplified according to the manufacturer’s protocol. The PCR program for amplification of the oxacillinase (OXA) genes OXA-23, OXA-24, and OXA-51 was conducted at 94 °C for 5 min, followed by 30 cycles at 94 °C for 25 s, 52 °C for 40 s, and 72 °C for 50 s. The program finished at 72 °C for 6 min. Gel electrophoresis, sequencing, and BLAST at National Center for Biotechnology Information (https://blast.ncbi.nlm.nih.gov/Blast.cgi, accessed on 27 October 2023) were performed. A sequence alignment similarity to the database’s accession of at least 99% was considered homologous. The Oxford scheme was matched using MLST analysis (http://pubmlst.org/abaumannii/, accessed on 7 November 2022).

### 2.2. Production of Monoclonal Antibodies

Male BALB/c mice weighing approximately 20 g were utilized. Sequential subcutaneous immunization of the mice with a sublethal dose (10^8^ CFU) of PDR *A. baumannii* was performed. The log-growth phase bacteria were mixed with complete Freund’s adjuvant (Sigma, Saint Louis, MO, USA, cat: F5881) at a 1:1 (*v*/*v*) ratio for the first injection, then replaced with the incomplete Freund’s adjuvant (Sigma, cat: F5506) in three subsequent booster doses administered at 2-week intervals. After 3 days of booster immunization, mice splenocytes with Sp2/0 cells were used for hybridoma cell fusion. The hybridoma cells were screened four times for consecutive monoclonal hybridomas. The isotypes of the mAbs were tested using ELISA with a secondary antibody of goat anti-mouse antibodies conjugated to horseradish peroxidase (IgG1, Abcam, Cambridge, UK, cat: ab97240; IgG2a, Abcam, cat: ab97245; IgG2b, Abcam, cat: ab97250; and IgG3, Abcam, cat: ab97260). The antibodies were diluted to 1:20,000 or 1:40,000. Unrelated antibodies were included as isotype controls. The unwound light and heavy chains of the monoclonal antibodies were verified via PAGE.

### 2.3. Whole-Cell ELISA

To assess the targets of mAb-specific binding to heterologous strains of MDR *A. baumannii*, 13 strains (Table 1) were cultured overnight to mid-log growth and washed twice in phosphate-buffered saline (PBS). The 50 µL *A. baumannii* samples were incubated in 96-well plates for 1 h at 37 °C. The same volume of 4% paraformaldehyde was added to the bacteria samples for 10 min. Blocking was conducted overnight at 4 °C with 5% non-fat milk. The mAbs (50 µg) were distributed in each well. Goat anti-mouse H&L (Abcam, cat: ab205719) and the TMB substrate chromogen (Beyotime, Shanghai, China, cat: P0209) were used to detect binding to the bacteria.

### 2.4. Western Blot

To verify the two mAbs recognized bacterial epitopes, log-phase bacteria were washed twice in phosphate buffer solution. Bacterial lysates were obtained by ultrasonic cell shredder (Scientz, Ningbo, China) in an ice bath. The unbroken bacteria were removed by 1000× *g* centrifugation. Suitable volumes of bacterial lysates were mixed with 5 × SDS-PAGE Protein Loading Buffer containing sodium dodecyl sulfate and dithiothreitol (GBCBIO Technologies, Guangzhou, China, cat: G3422), boiled at 100 °C for 5 min, separated on 10% SDS-polyacrylamide gels, and the protein sample was transferred onto a polyvinylidene fluoride (PVDF) membrane with a pore size of 0.45 µm. The PVDF was blocked overnight with 5% skim milk/0.1% Tween-20/PBS at 4 °C, and then incubated overnight with l ug/mL monoclonal antibody diluted by blocking buffer for 1 h. Goat anti-mouse IgG coupled with horseradish peroxidase was used for testing. Protein bands were visualized using a chemiluminescent ECL reagent (GBCBIO Technologies, cat: G3308) on the Bio Red^TM^ ChemiDocImaging System. ImageJ2 was used to compare the gray values of the respective lanes [27].

### 2.5. Co-Immunoprecipitation

To expand the scope of analysis beyond the constraints of traditional immunoblotting techniques, the immunoprecipitation method was performed. Outer membrane protein isolation was obtained from fresh bacterial culture medium by ultrasonic cell shredder (Scientz, China) in an ice bath, and the unbroken bacteria were removed by centrifugation. An Immunoprecipitation Kit (Beyotime, cat: P2197S) was used according to the manufacturer’s protocol. In summary, non-specific membrane proteins were precipitated using Protein A/G agarose gel conjugated with a normal IgG antibody, used as the internal component of the kit. mAb 1B5 or mAb 8E6 was added to the gel suspension at a ratio of 20:100 μL with the protein supernatant sample, incubated overnight at 4 °C on a rotary mixer, and the gel was collected and washed three times. The target protein was harvested by acid elution, and the protein concentration was determined by the BCA method.

### 2.6. LC-MS/MS Analysis

The targeted proteins obtained from co-immunoprecipitation were separated and identified using 10% SDS-PAGE, and the bands were collected for qualitative identification by LC-MS/MS protein spectrometry (Oebiotech, Shanghai, China). The database of blasting *Acinetobacter baumannii* 1419130 was obtained from UniProt [28].

Online database analyses were performed, antibody 3D structures were modeled by the automated LYRA program [29], antigen models were obtained from the SWISS-MODEL Repository database [30], and antibody–antigen docking was predicted by Cluspro 2.0 web servers [31].

### 2.7. In Vitro Complement Susceptibility and Phagocytosis Assay

As the first line of defense against invading microorganisms, neutrophils and complements play essential roles in host resistance against infection with *A. baumannii*. To measure complement susceptibility, log-phase *A. baumannii* were cultured overnight, washed twice with PBS, and resuspended to 4 × 10^6^ CFU/mL. The complement was collected from female guinea pigs, the bacteria and mAbs were added to 96-well cell culture plates for volume in a 50:25:25 μL ratio, and the complement was diluted 100-fold and inactivated at 56 °C for 30 min as a control. The 96-well cell culture plate was then incubated for 1 h at 37 °C after shaking at 150 rpm for 10 min. The bacterial burden was quantified by serially diluting bacterial cultures, and the CFU numbers were counted after 12 h.

HL-60 cell lines (CL-0110), obtained from Procell (Wuhan, China), were differentiated into neutrophil-like cells by culturing with 1.25% dimethyl sulfoxide (SIGMA-ALDRICH, D2650) for 5 days. The differentiated HL60 cells (dHL-60), complement, mAb, and bacteria were inoculated in 96-well cell culture plates, and the ratio of bacteria to dHL-60 cells was <200:1. The complement was heat-inactivated at 56 °C for 30 min as a control. The same treatment conditions were applied to the irrelevant isotype mAb. The component samples were centrifuged at 200 rpm for 20 min prior to activation. Then, the samples were cultivated at 37 °C and 5% CO_2_ for 45 min. The samples were then placed on ice for 20 min. Serial dilutions of cultures were plated on solid agar medium, and the colony counts were determined after 12 h.

### 2.8. Mouse Infection Model

Male BALB/c mice (between 8 and 10 weeks of age and weighing approximately 20 g) of a respiratory-pneumonia-associated oropharyngeal pneumonia model were used in this study. The mice were temporarily anesthetized using isoflurane, and 50 µL of fresh bacteria were applied to each nostril, allowing the bacteria to invade the lung tissue via reflexive aspiration. The mAbs were immediately administered via intraperitoneal injection (20 mg/kg). The mouse lung tissues were isolated, and fixed sections were stained to observe pathological changes. The animals were fed under appropriate conditions and maintained in an aerated environment.

### 2.9. Cytokine Assay

After 4 and 24 h of mice infected with SZ-AB57 *A. baumannii* (5 × 10^8^ cfu per mouse), the mice were humanely euthanized, and their lung tissues were collected and processed using a cryogenic grinder. The tissues were homogenized with grinding beads at a frequency of 70 Hz for 4 min at a temperature of 4 °C. Following homogenization, the samples were centrifuged at 10,000 rpm for 5 min at 4 °C, and the supernatant was collected and stored at −80 °C. Cytokine concentrations were quantified using five cytokine ELISA kits: Mouse IL-1β ELISA Kit (SEKM-0002), Mouse IL-6 ELISA Kit (SEKM-0007), Mouse IL-10 ELISA Kit (SEKM-0010), and Mouse TNF-α ELISA Kit (SEKM-0034), obtained from Solarbio (Beijing, China). The Mouse IL-17 ELISA Kit (MU30074) was obtained from Bioswamp (Wuhan, China), in strict accordance with the protocols outlined in the respective kit manuals.

### 2.10. Statistical Analysis

The data were tested for normality and then subjected to a one- or two-way analysis of variance (ANOVA) or unpaired *t*-tests. GraphPad Prism software (version 10) was used to create graphs. The graphical data are presented as mean and standard error or mean and standard deviation. Statistical significance was set at *p* < 0.05.

## 3. Results

### 3.1. MLST Serotype and Virulence Genes of Clinically Isolated A. baumannii

The strains used in this study may be the most prevalent in the region (Table 1). ST-208 strains accounted for 61.53% (8/13) of those isolated from patient sputum samples; ST-195 accounted for 30.77% (4/13); and ST-229 accounted for 7.69% (1/13). All bacterial strains obtained from Shenzhen were non-susceptible to all antimicrobial agent categories, including aminoglycosides, antipseudomonas carbapenems, antipseudomonas fluoroquinolones, and six other kinds of anti-microbial categories and agents [32], whereas the strains from Zhuhai were sensitive to ≥3 antimicrobial categories. The most common mechanism of carbapenem resistance was degradation by carbapenemase enzymes. The OXA-23 and OXA-51 subgroups were detected in all isolates (13/13).

### 3.2. Monoclonal Antibodies Obtained by Sequential Immunization Provide an Effectively Cross-Recognition Effect

Sequential immunization with low homologous serotype *A. baumannii* produced antiserum that broadly cross-bound three strains of *A. baumannii* clinal isolates (1 × 10^6^ CFU/well) (Figure 1). The maximum monitoring limit of SZ-AB57 and SZ-AB61, as quantified by immunoglobulin G in the serum of immunized mice, was found to be 10.24 million-fold greater than that observed in the control serum (*p* = 0.0083, *p* < 0.0001). Additionally, the maximum detection limit of SZ-AB22 was established to be 1.28 million-fold higher than that of the control serum (*p* = 0.0376).

### 3.3. Monoclonal Antibodies Obtained by Sequential Immunization Provide an Effective Cross-Opsonophagocytosis Effect

Two monoclonal antibodies, mAb 8E6 (IgG 2a) and mAb 1B5 (IgG 2a), were obtained by hybridoma technique, and the mechanism of antibacterial effect in vitro was explored by verifying the susceptibility to complement assay. The mAb 8E6 was verified to significantly clear SZ-AB61 in 25 ng per well compared to 3, 6, and 12 ng in a 100-fold diluted complement concentration range (Figure 2A, *p* = 0.0019). At the same concentration of complement, both mAbs significantly eliminated SZ-AB22 PDR *A. baumannii* (Figure 2B, *p* < 0.0001).

The bactericidal capability also appears in the opsonophagocytosis process associated with the clearance of bacteria by dHL-60 cells. The SZ-Ab57, SZ-Ab22, and SZ-AB61 strain loads decreased by approximately 66–75% after treatment with mAb 8E6 and mAb 1B5 (Figure 2C). Both mAbs killed the heterogeneous strains of the MLST serotypes ST-208 and ST-195 when combined with complement or dHL-60 cells at significant rates (Figure 3A). The median killing rate of mAb 8E6 when complement and dHL-60 cells were present was 61.51% (interquartile range [IQR]: 48.25–69.16%), and that of mAb 1B5 was 41.96% (IQR: 34.10–54.93%) (Figure 3B). When only dHL-60 cells were present, the killing rate of mAb 8E6 was 65.73% (IQR: 59.69–77.71%), and that of mAb 1B5 was 69.93% (IQR: 47.66–70.88%). Both mAbs significantly increased the ability of complement and dHL-60 cells to lyse *A. baumannii*, although there was no significant difference in the killing rate between the two mAbs under different conditions for MDR *A. baumannii*. The coefficients of variation for the killing rate of mAb 8E6 were 18.65% and 20.14%, while the coefficients of variation for mAb 1B5 were 32.55% and 32.52%. The antibacterial killing capacity of mAb 8E6 demonstrated less variability in comparison to mAb 1B5.

### 3.4. The mAbs against A. baumannii Infection through the Potentiation of Inflammatory Mediator Release

mAb 1B5 and mAb 8E6 showed better protection than the isotype mAb after infection with *A. baumannii* (Figure 4A). The mice administered mAbs returned to their average weight 1–2 days earlier than the control mice when infected with SZ-AB57 and SZ-AB61. The mAb 8E6 showed a similar protective effect to mAb 1B5.

The bacterial density of the mouse lung tissue between 4 h and 24 h after infection with SZ-AB57 revealed similar degrees of reduction (Figure 5A). The alveolar cavity of the antibody-treated group was filled with a significant number of inflammatory cells, including granulocytes and macrophages (Figure 4B). Few lymphocytes were detected in the control group. No significant difference regarding the number of granulocytes or macrophages was observed when the cells were treated with isotype mAb and PBS.

Upon measuring cytokine concentrations of 4 h and 24 h post-infection (Figure 5B), IL-10 levels in mice treated with mAb 8E6 were significantly elevated at the initial stage of infection (4 h, *p* = 0.019) compared to those in mice that received isotype mAb treatment. Both types of mAb-treated mice showed a significant IL-10 reduction in lung tissue post 24 h (*p* = 0.007, *p* = 0.007). Mice treated exclusively with mAb 8E6 exhibited increased levels of IL-6 in the lungs at the onset of infection (4 h, *p* = 0.041; 24 h, *p* = 0.038). An examination of TNF-α levels in the lungs indicated that only the mAb 8E6 treatment group displayed elevated cytokine levels post 4 h compared with mice given PBS (*p* = 0.036). And the mice treated with mAb 1B5 showed reduced TNF-α after 24 h (*p* = 0.049). No significant differences were detected in IL-1β and IL-17 levels between the two mAbs and the isotype mAb at the two time points assessed. Notably, IL-1β levels in the mAb 8E6 treatment group were significantly reduced 24 h post-infection (*p* = 0.044).

### 3.5. Cross-Recognition Bacterial Epitope Identification

Regarding the whole-cell ELISA results, both mAbs recognized *A. baumannii* intact cells obtained from two areas of south China, which was significantly more than in the control mAb (Figure 6A). The recognition of mAb 8E6 and mAb 1B5 with the bacterial surface was significantly diminished from the *A. baumannii* after treatment with proteinase K (Figure 6B), and the contrast of gray values was significantly different (Figure 6C, *p* = 0.0196). The results of protein identification by co-immunoprecipitation showed that the highest score of mAbs against SZ-AB01 and SZ-AB57 *A. baumannii* was ATP synthase (Figure 7).

Regarding antibody–antigen 3D prediction, the non-CDR regions of the antibody model are shielded in the 3D antibody–antigen docking process. The number and energy parameter sets of multiple models based on the fast Fourier transform correlation method are effective bases for screening the docking structure closest to the native amino acid structure (Figure 8). mAb 1B5 was far more present on the clusters than the 136 matched-site amino acids; the energy consumption was −358.8; the lowest energy consumption was −370.2. After excluding the non-accessible region of mAb 8E6 from the docking model, a docking model with 26 amino-acid-binding cluster centers was included, and the center binding potential energy consumption and the minimum binding energy consumption were both −340.1.

## 4. Discussion

Passive immunization is a promising therapeutic option against drug-resistant bacteria, with a variety of mature products. Human mAb products against multi-antibiotic-resistant bacteria, such as *Pseudomonas aeruginosa*, *Staphylococcus aureus*, *Escherichia coli*, and *Klebsiella pneumoniae*, have been developed in recent years [33]. However, no vaccine candidates for *A. baumannii* have been reported. We created two mAbs (mAb 8E6 and mAb 1B5), which specify against *A. baumannii* by enhancing the host immune response to kill bacteria by cross-recognizing the ATP synthase of bacteria.

*A. baumannii* is a widely isolated pathogen in clinical intensive care units, most of which occur in critically ill patients and are associated with very low survival, and fewer than 30% of critically ill patients can survive to discharge after infection with *A baumannii* [34]. In the face of pulmonary infection or combined bloodstream infection, immune dysfunction is one of the most important risk factors, and a disrupted balance of complement-dependent cytotoxic function makes the host more vulnerable to *A. baumannii* infection, which invades the bloodstream to induce more severe disease processes by escaping the complement system or by NETs released from neutrophils [35]. mAb 8E6 and mAb 1B5 specifically recognize binding MDR *A. baumannii* surface molecules, enhance the susceptibility of low concentrations of complement to *A. baumannii* in vitro, and almost completely clear bacteria. Invading the host immune system is necessary for the survival of bacteria after infection, as blocking disease development process progression is one of the most effective treatment strategies. However, we only performed in vitro experiments and did not identify immunocompromised mouse cases of more immune dysfunction, and the efficacy of mAbs was investigated in *A. baumannii* strains that exhibited a lack of complement resistance strains. Further comparison of the effects of sequential immunization strategies with other immunization strategies is limited, particularly regarding strategies involving a single strain or multiple strains administered simultaneously. Furthermore, it is essential to assess the immune responses associated with the mechanisms of sequential immunization strategies.

We observed the in vivo microstructural changes in mice treated with two antibodies in protection against *A. baumannii*. Mice administered with antibodies had severe inflammatory cell infiltration in the lung structures, smaller alveolar spaces, and alveolar edema in the early stages of *A. baumannii* infection at high doses. Even after 24 h, this inflammatory change gradually diminished restoring part of the alveolar structure. This process may have contributed to the restoration of normal body weight, which is why mice treated with mAb 8E6 and mAb 1B5 regained body weight earlier than those treated with control treatments. However, this may also be accompanied by a potentially excessive immune response, influenced by a balance between pro- and anti-inflammatory. Excessively positive pro-inflammatory responses or inadequate anti-inflammatory responses can lead to disturbances in immune response homeostasis [36]. Appropriate inflammatory modulation may play a critical role in the effect of monoclonal antibodies against *A. baumannii* infection. Neutrophil-derived interleukin-10 plays a key role in bacterial clearance by the immune system after the *A. baumannii* infection of mice, while TNF-α produced by macrophages plays an opposite role in monoclonal antibody therapy; for monoclonal antibodies to enhance microbial clearance by opsonophagocytosis, clinical efficacy requires the regulation of pro- and anti-inflammatory cytokines [37]. During the initial phase of infection, we detected elevated levels of IL-10 in mice that received mAb 1B5 and 8E6. IL-17 intensified the inflammatory response of the host to *A. baumannii* infection by suppressing the phagocytic activity of neutrophils in the initial stages [38]. Nonetheless, our findings revealed no notable variations in IL-17 levels when comparing the different groups. This distinct modulation could represent a promising therapeutic approach to managing the prognosis of *A. baumannii*-induced pneumonia. The modulation of cytokines has been shown to be effective in infections with other pathogens, such as anti-interleukin-6 monoclonal antibodies, which play a beneficial role in the treatment of cytokine storms caused by COVID-19 [39]. At the same time, antibody–cytokine fusion protein technology can not only carry cytokine receptors but also provide FcγR to mediate antibody-dependent cell-mediated cytotoxicity. However, the functional mechanism and safety of cytokines regulated against *A. baumannii* infection are still not well understood, and further studies are still needed.

ATP synthase is essential for obligate aerobes, and we identified two antibodies as targets for the cross-recognition of *A. baumannii* by co-immunoprecipitating target proteins from bacterial lysate. F1Fo-ATP synthase is a large membrane-embedded macromolecular complex that uses the energy of proton motive force (pmf) to synthesize ATP through a unique rotational mechanism [40]. The antibody–antigen complex prediction points to the C subunit as well as the A subunit in the Fo complex, and a/c10 is conserved in the Acinetobacter family. This is a substantial advantage in distinguishing other bacteria or pathogens, and the unique a/c10 *A. baumannii* interface is considered a main target for the development of highly specific inhibitors [41]. Because the highest affinity binding sites C_10_ ring and a subunit were found at the interface between the C subunit of ATP synthase, the ATP synthase inhibitor bedaquiline (BDQ) was approved by the US Food and Drug Administration in 2012 to be included in the treatment regimen for pulmonary drug-resistant tuberculosis [42]. BDQ remains by far the only ATP synthase inhibitor approved for the treatment of bacterial infections. BDQ primarily interacts with various sites within the ATP synthase of Mycobacterium tuberculosis. This interaction inhibits the rotation of the c ring, obstructing ATP synthesis and ultimately exerting a bactericidal effect [43]. However, the important indicator of antibody science is the accessibility of the target antigen, and this F1Fo-ATP synthase is usually predicted to be located on the bacterial plasma membrane [44]. The abundance of F1Fo-ATP synthase in the bacterial outer membrane is unknown, and although in-depth studies on how antibodies recognize this protein to exert antibody-dependent cytotoxicity are still required, F1Fo-ATP synthase remains a potential target for antibacterial strategies and antibiotic combination therapies.

## 5. Conclusions

mAb 8E6 and mAb 1B5 provided protection to mice against infection with *A. baumannii*, as distinguished through our investigation into samples from southern China by cross-recognition of the ATP synthase complex. This targeting enhances the killing ability of the complement system and neutrophils and provides opsonophagocytosis effects in mice. These results suggest that the preparation of antibodies against *A. baumannii* by sequential immunization strategy tailored according to the dominant strain in different geographical areas is a potentially effective route.

## Figures and Tables

**Figure 1 vaccines-12-01120-f001:**
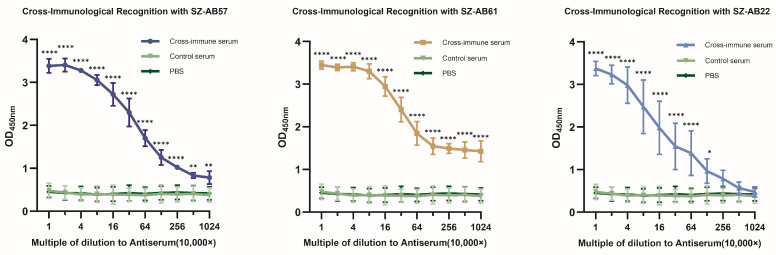
Each round of the sequential immunization strategy utilizes one of the three strains of PDR *A. baumannii*, eliciting strong cross-immune responses from sera that are generated through multiple rounds of immunization. The data are presented as the mean ± standard deviation (SD). A two-way ANOVA was conducted; *, *p* < 0.05; **, *p* < 0.01; ****, *p* < 0. 0001 versus isotype mAb group. There was no statistically significant difference in the OD450 nm values among the three strains of *A. baumannii* when comparing the two distinct continuous detection gradients.

**Figure 2 vaccines-12-01120-f002:**
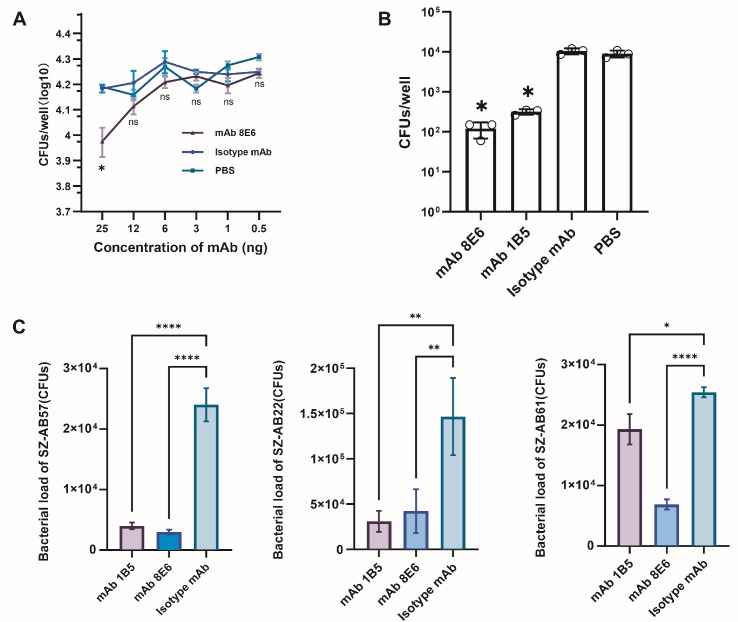
(**A**) The mAbs-mediated complement and neutrophil bactericidal effects. The mAb 8E6-mediated complement susceptibility to SZ-Ab61 (ST-229) *A. baumannii* by guinea pig serum of 100 times dilution; the dose of antibody was 0.5–25 ng. (**B**) The bactericidal ability of SZ-Ab22 (ST-195) of the complement enhanced by mAb 8E6 (25 ng/well) also appeared in guinea pig serum diluted in 1:100. (**C**) Both mAbs showed bactericidal ability of the complement and dHL-60 cells against immune strains. Significant differences were detected using the one-way or two-way ANOVA test; *, *p* < 0.05; **, *p* < 0.01; ****, *p* < 0.0001; ns, not significant versus isotype mAb group. All tests were repeated three times.

**Figure 3 vaccines-12-01120-f003:**
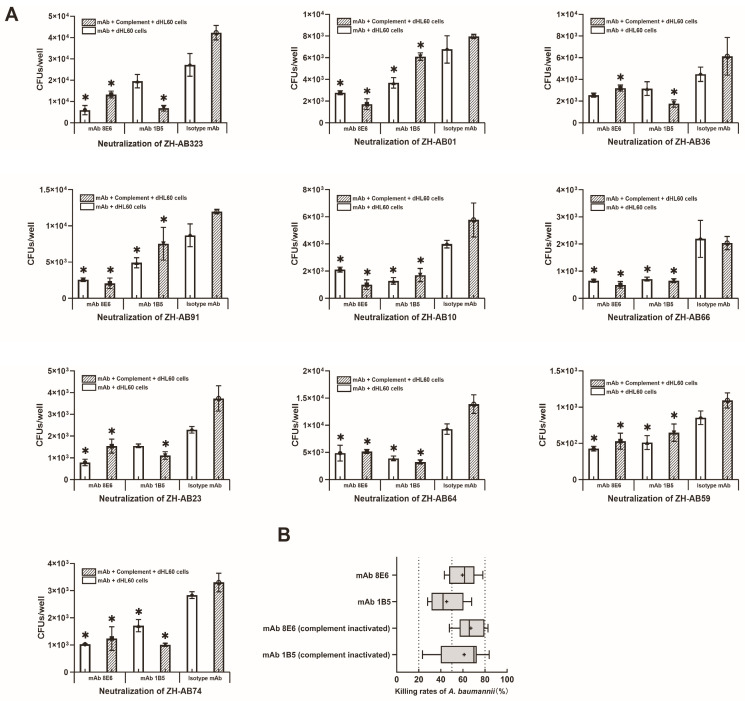
(**A**) The mAb 1B5- and mAb 8E6-mediated killing heterogenous ST-208 and ST-195 *A. baumannii* with specificity when combined with guinea pig complement and dHL-60 cells. Each well was consistently replicated in three wells within each experimental cohort; *, *p* < 0.05 versus isotype mAb group by Tukey’s post hoc test. The complement was diluted 3000 times to achieve a non-specific killing rate (NSK) < 25% in the PBS control. The NSK was calculated as NSK = [1 − (number of colonies in complement control group/number of colonies in inactivated complement control group)] × 100%. (**B**) The bactericidal rates of mAb 8E6 and mAb 1B5 against 13 strains of antibiotic-resistance *A. baumannii* were combined, and the median values were marked with “+”, no statistically significant differences (*p* > 0.05) were observed in the rates of bacterial killing among mAb 8E6 and mAb 1B5.

**Figure 4 vaccines-12-01120-f004:**
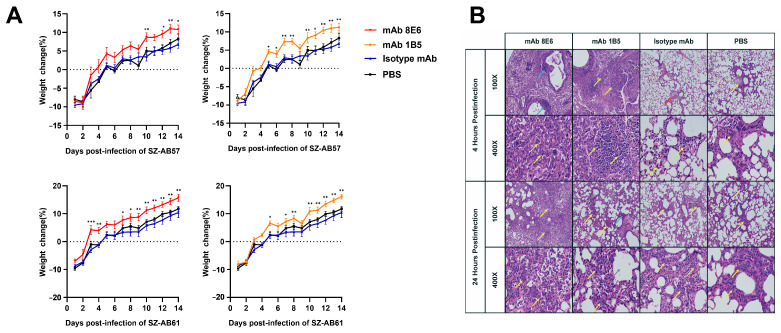
The mAbs enhancing the release of immune substances. (**A**) Mice were infected with ST-208 and ST-229 *A. baumannii* via nasal intubation drip, and then immediately injected into the mice (n = 10). All mice survived the study period. Significant differences were detected using the Tukey’s Honestly Significant Difference test; *, *p* < 0.05; **, *p* < 0.01; ***, *p* < 0.001 versus isotype mAb group. All tests were repeated three times. (**B**) The microscopic architecture of the lungs in mice inoculated with PDR *A. baumannii* was scrutinized following mAbs protection. The lung tissue was separated at a specified time point and fixed with 4% paraformaldehyde. The rest of the lung tissue homogenates were 10-fold diluted and counted. The sections were stained with hematoxylin and eosin stain and observed under 100× and 400× magnification. More inflammatory cells were observed in the immunized mice during the early stages of the disease. The normal tissue structure recovered more quickly in the immunized mice than in the control group. Inflammatory cells are indicated by yellow pointers, while small focal infiltrations of lymphocytes are represented by blue pointers. Dot infiltrations of lymphocytes and granulocytes are denoted by red pointers, and compensatory increases in the surrounding alveoli are illustrated with gray pointers.

**Figure 5 vaccines-12-01120-f005:**
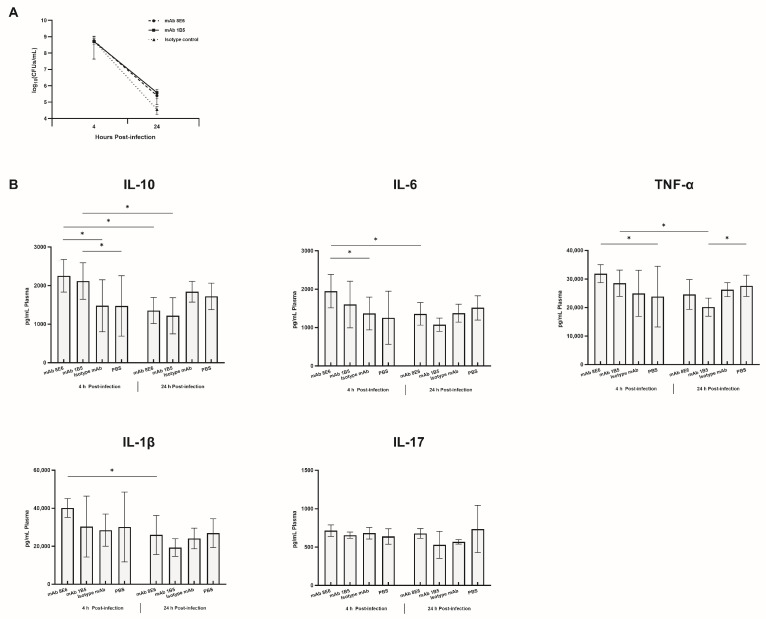
Mice were inoculated with *A. baumannii* and subsequently treated with either an antibody or a control treatment (n = 5). Lung tissues were harvested at 4 and 24 h post-infection to assess bacterial load and cytokine levels. (**A**) No significant disparity in pulmonary bacterial burden was observed between the groups treated with the two distinct antibodies and the control group. (**B**) The cytokine levels in the lungs of mice administered mAbs exhibited significant variation when compared to the control group; *, *p* < 0.05 compared to the isotype mAb or the PBS, as determined by the least significant difference (LSD) test.

**Figure 6 vaccines-12-01120-f006:**
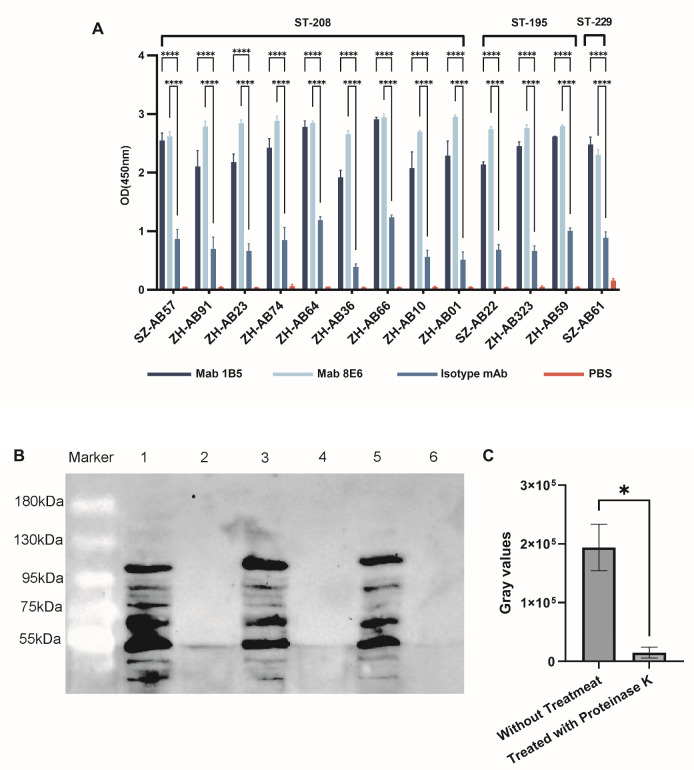
(**A**) Antibodies mediated cross-recognize heterogenous *A. baumannii* identification. The mAb 8E6 and mAb 1B5 specifically target *A. baumannii*, as determined using whole-cell ELISA, which was not observed in the isotype mAb; ****, *p* < 0.0001, versus isotype mAb group by Dunnett’s *t*-test. (**B**) Western blot lanes 1, 3, and 5 represent bacterial lysate protein samples SZ-AB57, SZ-AB22, and SZ-AB61, respectively. Meanwhile, lanes 2, 4, and 6 were the lysates of these three bacteria treated with protease K separately. (**C**) The gray values obtained from the two methods were aggregated and analyzed using a paired *t*-test; *, *p* < 0.05.

**Figure 7 vaccines-12-01120-f007:**
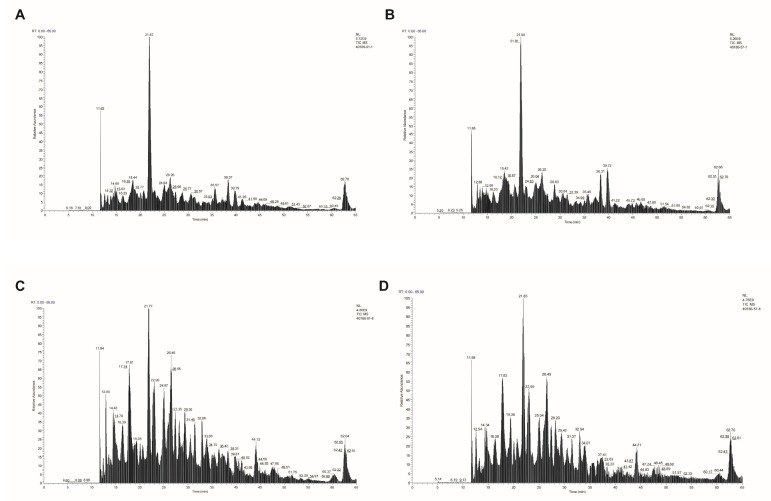
LC-MS/MS protein spectrometry results. X-axis represents *m*/*z* (mass-to-charge ratio); Y-axis represents relative abundance. The results of co-immunoprecipitation of mAb 1B5 with strain ZH-AB01 are shown in (**A**), the mAb 1B5 with strain SZ-AB57 are shown in (**B**); the co-immunoprecipitation of mAb 8E6 with strain ZH-AB01 are shown in (**C**), the mAb 8E6 with strain SZ-AB57 are shown in (**D**).

**Figure 8 vaccines-12-01120-f008:**
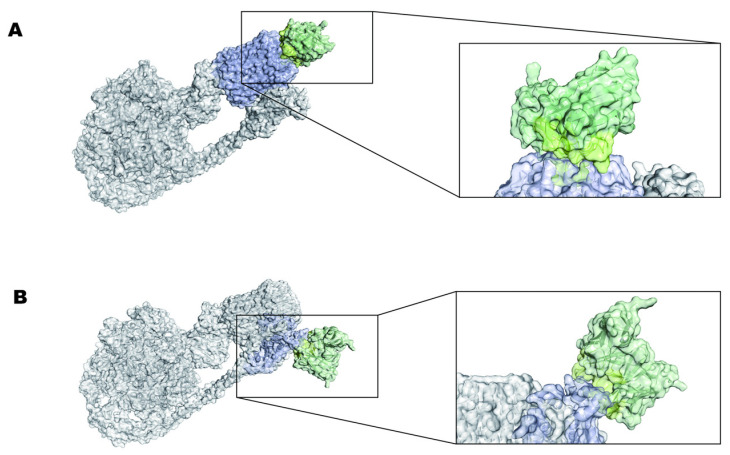
3D structure of antibody–antigen combination forecast. The light purple color represents the position of the antigen subunit in the macromolecular ATP synthase. Green region is the antibody variable region structure simulated by the LYRA database, and the yellow region is the complementarity determining region identification combined with antigen epitope prediction. (**A**) The mAb 1B5 adheres to the subunit a. (**B**) mAb 8E6 adheres to the c ring.

**Table 1 vaccines-12-01120-t001:** The allelic genes were assigned using the Oxford multi-locus sequence typing scheme.

Isolate Strains	Isolate Source	Allelic Distribution *	STs	Susceptibility	Global Clonal	Year/Country	Virulence Genes		
							OXA-23	OXA-24	OXA-51
SZ-AB57	Sputum	1, 3, 3, 2, 2, 97, 3	208	PDR	GC2	2021/Shenzhen, China	+	−	+
SZ-AB61	Sputum	1, 15, 2, 28, 1, 107, 32	229	PDR	GC1	2021/Shenzhen, China	+	−	+
SZ-AB22	Sputum	1, 3, 3, 2, 2, 96, 3	195	PDR	GC2	2021/Shenzhen, China	+	−	+
ZH-AB74	Sputum	1, 3, 3, 2, 2, 97, 3	208	MDR	GC2	2022/Zhuhai, China	+	−	+
ZH-AB64	Sputum	1, 3, 3, 2, 2, 97, 3	208	MDR	GC2	2022/Zhuhai, China	+	−	+
ZH-AB59	Sputum	1, 3, 3, 2, 2, 96, 3	195	MDR	GC2	2022/Zhuhai, China	+	−	+
ZH-AB91	Sputum	1, 3, 3, 2, 2, 97, 3	208	MDR	GC2	2022/Zhuhai, China	+	−	+
ZH-AB23	Sputum	1, 3, 3, 2, 2, 97, 3	208	MDR	GC2	2022/Zhuhai, China	+	−	+
ZH-AB323	Sputum	1, 3, 3, 2, 2, 96, 3	195	MDR	GC2	2022/Zhuhai, China	+	−	+
ZH-AB01	Sputum	1, 3, 3, 2, 2, 96, 3	195	MDR	GC2	2022/Zhuhai, China	+	−	+
ZH-AB10	Drainage fluid	1, 3, 3, 2, 2, 97, 3	208	MDR	GC2	2022/Zhuhai, China	+	−	+
ZH-AB36	Sputum	1, 3, 3, 2, 2, 97, 3	208	MDR	GC2	2022/Zhuhai, China	+	−	+
ZH-AB66	Sputum	1, 3, 3, 2, 2, 97, 3	208	MDR	GC2	2022/Zhuhai, China	+	−	+

* The allelic distribution of the housekeeping genes gltA, gyrB, gdhB, recA, cpn60, gpi, and rpoD. Capsule serotypes in the strains have not been defined. A ‘+’ indicates a positive amplification result, while a ‘−’ indicates the absence of a BLAST result.

## Data Availability

All relevant data are presented within the article.

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
