# Peer review of "Sequential Immune Acquisition of Monoclonal Antibodies Enhances Phagocytosis of Acinetobacter baumannii by Recognizing ATP Synthase"

_vaccines, 2024, doi:10.3390/vaccines12101120_

Round 1

Reviewer 1 Report (Previous Reviewer 1)

Comments and Suggestions for Authors

The authors have adequately addressed all my previous comments.

Comments on the Quality of English Language

Please replace bacterial density with bacterial burden.

Author Response

The authors have adequately addressed all my previous comments.

Response: Many thanks. Have a lovely day.

Reviewer 2 Report (Previous Reviewer 4)

Comments and Suggestions for Authors

I have gone through the resubmitted manuscript. The following comments were still not addressed aby the authors:

1. Only three mice were included in each group, this is an insufficient number. 

2. This study mentions the possibility of excessive immune responses. This aspect needs more in depth investigation to ensure the safety and efficacy of the therapy. 

3.Although the study shows the binding of Mabs to ATP synthase, it points limited mechanistic insight into how this binding translates into increased bacterial clearance. Further explanation of pathways could strengthen the study.

4. Authors have not explained other vaccine strategies to combat drug-resistant A. baumannii infections.

Round 2

Reviewer 2 Report (Previous Reviewer 4)

Comments and Suggestions for Authors

I have gone through the authors` response. I am satisfied with the response.

This manuscript is a resubmission of an earlier submission. The following is a list of the peer review reports and author responses from that submission.

Round 1

Reviewer 1 Report

Comments and Suggestions for Authors

In this manuscript, the authors report the generation of monoclonal antibodies from mice sequentially immunized with 3 Acinetobacter baumannii strains. The authors further characterize two neutralization antibodies, Mab 8E6 and Mab 1B5, and identify the bacterial ATP synthase complex as their putative binding epitopes.  Protective efficacy studies reveal that these two Mabs can enhance opsonophagocytic bacterial killing against multiple A. baumannii strains.

A. baumannii is a major cause of nosocomial infections. The rapid emergence of multi-drug resistant (MDR) Ab infection is increasingly difficulty to treat, thus, development of immunotherapy including Mab described in this manuscript is an urgent need to control this bacterial disease. However, there are major shortcomings that reduce the scientific impact of this report.

 (1) The authors seem to imply (title and introduction) that sequential immunization with 3 A. baumannii leads to cross antibody (Mab 8E6 and Mab 1B5) reactivity with broad range bacterial strains. Although not fully understand the immune mechanisms of sequential immunization, most likely the difference in antigen dominance of each immunization and generation of diverse polyclonal antibodies via affinity maturation, but not a specific monoclonal antibody, are responsible for expanding overall antibody cross reactivity with multiple bacterial strains. Additionally, the authors did not provide evidence that combined 3 stain vaccination fail to generate Mab 8E6 and Mab 1B5. Furthermore, if ATP synthase complex contains Mab 8E6 and Mab 1B5 binding epitopes, one should ask how conserve are these epitopes among cross-reacted A. baumannii strains.

 (2) It is well documented that some A. baumannii strains are complement-resistant, the others are susceptible to complement killing. The authors have not use complement susceptible strains to access Mab’s bactericidal activity.

 (3) The bacterial activity of Mabs against SZ-Ab61 strain is independent of complement (Fig. 2B). Would the Mab “neutralization” activity come from ATP synthase inhibition? Experiments can be designed to address this important question (e.g. Mab mediated inhibition of ATP synthase enzymatic activity, and bacterial growth curve assessment in the presence of Mabs).

 Other minor comments:

 (4) The authors need to keep in mind the difference between antibody neutralization and opsonization activity in data interpretation. 

(5) Reference 1: This report does not mention inappropriate or unnecessary use of antibiotics. Also, the reference link is not working. 

(6) Lines 58: Replace (MAC)-mediated bacterial “cleavage” with bacterial “killing”. 

(7) Line 61: Rephrase “The germicidal range of supporting evidence for Mab remains limited.” to clarify what the “range” means. 

(8) Lines 199-200: Change bacterial “density” to bacterial “burden” and CFU “amounts” to CFU “numbers”. 

(9) Lines 202-204:  Please provide reference and clarify “human neutrophils” as neutrophil-like cells derived from human leukemia HL-60 cell line.  

(10) Line 233: Please define “all antimicrobials”. 

(11) Fig 1. What is the sequential immunization regimen? What is the antigen coated in the ELISA? Does single strain vaccination provide cross strain reactive antibody?  

(12) Fig. 1D makes no sense. Are these 5 indicated samples mouse sera or plate coated antigens? 

(13) Line 254, Fig 2 Results are not obtained from in vivo experiments. 

(14) Fig. 3K. what is the bacterial strain used in this study? 

(15) Line 278, how are stability and affinity measured? 

(16) Line 300: change “continuously dilution” to xx-fold serial dilution.

 (17) Fig 4F. It seems more inflammation was observed in Mab treated mice. How about other histopathology? Does inflammation induced by Mab treatment cause more lung damage than mock treatment?  

(18) Fig 5B, is it MAB 8E6 or Mab 1B5? Why so many proteins are reacted with monoclonal antibody? Does MAB 8E6 or Mab 1B5 react with the same proteins?

 (19) Lines 367-370. There is no evidence backing up the claim that MAB 8E6 and Mab 1B5 activate MAC-killing.

Comments on the Quality of English Language

Please see comments about proper usage of some common microbiology and immunology terms.

Reviewer 2 Report

Comments and Suggestions for Authors

Title: Sequential immune acquisition of neutralizing monoclonal antibodies enhances phagocytosis of Acinetobacter baumannii by recognizing ATP synthase.

Manuscript ID: 3099694

I recommended that manuscript could be accepted with MINOR MODIFICATIONS based on following GENERAL SUGGESTIONS:

-          Specify how the A. baumannii isolates were identified or added a reference.

-          In figure 1, almost in ´D´ statistical differences should be added.

-          In the legend of figure 3, 4, 5, and 7, specify the different letters included in each section.

-          Eliminate (A. baumannii) in abstract, write complete name at start a sentence and always is in cursive. Please, review in complete manuscript.

-          Review typographic mistakes and English language and grammar.

Reviewer 3 Report

Comments and Suggestions for Authors

In general terms, the study is well presented, the introduction addresses the health problem with this bacterial species.

The methodology is well written. However, considering the compartmentalized infection in the respiratory tract, wouldn't it be more appropriate after the experimental infection to also apply monoclonal antibodies via the respiratory route?

Considering the nature of the study, I missed the measurements of pro-inflammatory and immunomodulatory cytokines, as they would greatly enrich the results.

A peculiar situation is that the efficiency of these antibodies are directly related to local bacterial strains, which would certainly make the development of a unique product difficult.

Reviewer 4 Report

Comments and Suggestions for Authors

The present manuscript entitled "Sequential immune acquisition of neutralizing monoclonal antibodies enhances phagocytosis of Acinetobacter baumannii by recognizing ATP synthase" addresses an important healthcare issue, focusing on multi-drug resistant Acinetobacter baumannii. This study includes a novel strategy using sequential immunization to produce monoclonal antibodies that target ATP synthase of the pathogen. This approach could potentially overcome the limitations of single antigen based vaccines. However, there are certain points that should be taken into consideration in order to make this manuscript more effective.

1. In abstract section, some important quantitative parameters should be included.

2. Although the study shows the binding of Mabs to ATP synthase, it points limited mechanistic insight into how this binding translates into increased bacterial clearance. Further explanation of pathways could strengthen the study.

3. Only three mice were included in each group, this is an insufficient number.

4. This study mentions the possibility of excessive immune responses. This aspect needs more in depth investigation to ensure the safety and efficacy of the therapy. 

5. Include some important recent references related to protection studies. 

Safety and Prophylactic Efficacy of Liposome-Based Vaccine against the Drug-Resistant Acinetobacter baumannii in Mice. Pharmaceutics. 2022 Jun 27;14(7):1357.